# Blood-Based Immune Protein Markers of Disease Progression in Murine Models of Acute and Chronic Inflammatory Bowel Disease

**DOI:** 10.3390/biomedicines11010140

**Published:** 2023-01-05

**Authors:** Tyler Milston Renner, Gerard Agbayani, Renu Dudani, Michael J. McCluskie, Bassel Akache

**Affiliations:** National Research Council Canada, Human Health Therapeutics, Ottawa, ON K1A 0R6, Canada

**Keywords:** inflammatory bowel disease (IBD), colitis, biomarker, protein marker, cytokines, chemokines, dextran sulphate sodium (DSS), T cell transfer

## Abstract

Inflammatory bowel disease (IBD) is a chronic ailment afflicting millions of people worldwide, with the majority of recognized cases within industrialized countries. The impacts of IBD at the individual level are long-lasting with few effective treatments available, resulting in a large burden on the health care system. A number of existing animal models are utilized to evaluate novel treatment strategies. Two commonly used models are (1) acute colitis mediated by dextran sulphate sodium (DSS) treatment of wild-type mice and (2) chronic colitis mediated by the transfer of proinflammatory T cells into immunodeficient mice. Despite the wide use of these particular systems to evaluate IBD therapeutics, the typical readouts of clinical disease progression vary depending on the model used, which may be reflective of mechanistic differences of disease induction. The most reliable indicator of disease in both models remains intestinal damage which is typically evaluated upon experimental endpoint. Herein, we evaluated the expression profile of a panel of cytokines and chemokines in both DSS and T cell transfer models in an effort to identify a number of inflammatory markers in the blood that could serve as reliable indicators of the relative disease state. Out of the panel of 25 markers tested, 6 showed statistically significant shifts with the DSS model, compared to 11 in the T cell transfer model with IL-6, IL-13, IL-22, TNF-α and IFN-γ being common markers of disease in both models. Our data highlights biological differences between animal models of IBD and helps to guide future studies when selecting efficacy readouts during the evaluation of experimental IBD therapeutics.

## 1. Introduction

Incidence rates for inflammatory bowel disease (IBD), such as Crohn’s disease or ulcerative colitis, are rapidly increasing around the globe, especially in industrialized countries [1]. Not only does this have a huge impact on the patients’ quality of life, but as incurable chronic diseases they place a high burden on the health care system [2]. Typical treatments include the use of antibiotics to eliminate gut microbiota, dietary regulation, administration of corticosteroids/immunosuppressants and in severe situations, surgical removal of severely inflamed intestinal regions [3]. The side effects and limited success associated with these treatment options have led to the development of several biologics, but this has unfortunately not led to an overall improvement on health care costs [4,5,6]. For example, Tumor Necrosis Factor alpha (TNF-α) inhibitors have been used as a successful therapy for a fraction of IBD patients; however, the majority of recipients either have no response or exhibit a waning response over the course of treatment [7]. Similarly, antibodies targeting the integrins responsible for the immune cell trafficking to the intestinal tract have been used in an attempt to reduce inflammation but with limited success [8,9]. Given the significant percentage of patients that do not respond to currently approved treatments for IBD, a need for novel effective therapies exists.

Prior to entering the clinic, novel therapies are first evaluated within animal models of disease. There have been several reported mouse models of colitis that partially represent the human phenotype. Two widely accepted examples are the dextran sulphate sodium (DSS) and the T cell transfer model in mice [10]. Addition of DSS to drinking water disrupts the intestinal epithelial barrier in a dose-dependent manner, resulting in bacterial translocation of the gut microbiota into the lamina propria where they induce activation of the immune response [11]. The resulting inflammation triggers damage to the intestinal tissue, leading the mice to display a variety of IBD disease-like symptoms. By adjusting the timeline of exposure to DSS, mice can exhibit an acute or chronic-like state of disease that is mostly mediated by the innate immune response [11]. The T cell transfer model involves isolation of naïve proinflammatory T cells from healthy donor mice to be injected into immunodeficient mice of the same genetic background [12]. Due to the absence of regulatory T cells in the recipient mice, autoimmune reactions can develop with those in the intestinal tract being most prominent due to the constant interaction with gut microbiota [12]. Side-by-side, these two models independently capture the roles of the innate and adaptive immune responses in experimental models of colitis.

Typically, these models of colitis are evaluated for clinical signs of disease on a semi-quantitative scale of disease activity index (DAI), which incorporates weight loss, fecal blood occurrences and fecal consistency [13]. Other endpoint analyses include measurement of the length and/or weight of the colon, followed by histopathological scoring [11,12,13]. Other commonly used approaches include the measurement of cytokine/chemokine expression through qRT-PCR or ELISA of either serum or afflicted tissue, which requires a predetermination of the cytokines of interest [14,15,16,17]. Though multiplexing with techniques such as cytokine bead array assays are less common due to their increased cost, assessment of certain inflammatory markers (i.e., TNF-α, IFN-γ, IL-6, IL-17A, etc.) within the serum or afflicted tissues has been incorporated in some studies [18,19,20]. These types of analyses may be more capable of revealing therapeutic impacts than clinical observations given the time lapse between inflammatory reduction and tissue repair, an example being the dexamethasone treatment of mice with T-cell-transfer-mediated IBD [19].

Herein, we evaluated the systemic inflammatory profiles of two commonly employed murine models of inflammatory bowel disease: DSS-mediated acute colitis and chronic colitis induced by T cell transfer. The clinical and post-mortem analyses illustrated the typical findings for both models with regard to weight loss, histopathological scoring, and colon measurements. Using a multiplexed cytokine/chemokine assay, the levels of 25 inflammatory cytokines or chemokines were measured in the serum of these mice, identifying a number of markers with some interesting differences between the models. For example, IL-6, IL-13, IL-22, TNF-α and IFN-γ were upregulated in both models of disease, but levels of the latter two were much higher in the T cell transfer model at the time points tested. Overall, this study allows for the rational selection of serum-based markers for the evaluation of therapeutics in IBD models and highlights key biological differences between these two commonly used mouse models of IBD. 

## 2. Materials and Methods

### 2.1. Animals

Animals were maintained at the small animal facility of the National Research Council Canada (NRC) in Ottawa, Canada in accordance with the guidelines of the Canadian Council on Animal Care. All procedures performed on animals in this study were approved by our Institutional Review Board (NRC Human Health Therapeutics Animal Care Committee) and covered under animal use protocol 2020.01. All experiments were carried out in accordance with the ARRIVE guidelines.

### 2.2. DSS-Induced Mouse Model

Male (5–6 weeks old) and female (6–8 weeks old) C57BL/6 mice were obtained from Charles River Laboratories (Saint-Constant, QC, Canada) and housed 5 per cage. Animals were assigned into 5 groups, each containing 5 male and 5 female mice. Dextran sulphate sodium (DSS) with a mean molecular weight of 40 kDa and a 16–19% degree of sulphation was obtained from TdB Labs (Uppsala, Sweden). DSS was dissolved in drinking water to obtain a final mass/volume (*m*/*v*) ratio of 2–5% as indicated in the figures. Control mice received DSS-free water. Mice were provided with the water solutions ad libitum for 7 days. To minimize concerns over potential instability of DSS while in solution over time, drinking water solutions were replaced at 2 day intervals for the duration of the experiment. Mice and feces were visually monitored daily for clinical signs of disease similarly to a previously described method [13]. With a maximum score of 12, disease activity index (DAI) was scored according to Table 1. At Day 7, serum was collected for analysis of inflammatory markers as described below and then animals were euthanized prior to extraction of the colon. The colon was measured for its length. Colon weight was not noted, as it is typically reflective of the reduced length in acute colitis [11]. Feces was flushed from the colon, and the colon was fixed in buffered paraffin within a pinwheel orientation before Hematoxylin and eosin (H&E) staining and histopathological scoring according to Table 2 for a maximum score of 10.

### 2.3. T Cell Transfer Mouse Model

Male wild-type mice (6–8 weeks old C57BL/6), Rag1 -/- and Rag2 -/- (both 5–6 weeks old) were obtained from Jackson Laboratory (Bar Harbor, ME, USA) and housed individually. Spleens from the wild-type mice were processed into a single-cell suspension, magnetically isolated for CD4+ T cells via negative selection (StemCell Technologies, Vancouver, BC, Canada) and stained with APC-H7 anti-CD4 clone GK1.5, BV650 anti-CD25 clone PC61 and BV421 CD45RB clone 16A (BD Biosciences, Mississauga, ON, Canada). CD4+ CD25^low^CD45RB^High^ were sorted by fluorescence-activated cell sorting (FACS) on a MoFlo^®^ Astrios EQ (Beckman Coulter, Brea, CA, USA). The sorted cells were injected intraperitoneally (i.p.) (5 × 10^5^ cells in 100 µL PBS per animal) into 5 Rag1 -/- and 5 Rag2 -/- mice, which are both completely lacking in mature T and B cells. Mice and feces were monitored weekly for clinical signs of disease for the first 4 weeks, then monitoring frequency was increased to 2–3 times weekly until the animals either reached a humane endpoint (i.e., loss of more than 20% of initial body weight; moribund appearance and/or gross rectal bleeding) or the end of study period (64 days). A total of 4 mice (2 Rag1 -/- and 2 Rag2 -/-) were euthanized or found dead prior to the end of the study period, 1 of which died unexpectedly 4 weeks following transfer of T cells prior to exhibition of any disease symptoms and was therefore excluded from any analysis. 

At endpoint, serum was collected for analysis of inflammatory markers as described below prior to extraction of the colon and spleen. Both organs were measured for weight, but only the colon was measured for length. Feces were flushed and the colon was fixed within a pinwheel orientation before H&E staining and histopathological scoring as described above. Serum was also collected on Day 35 for analysis of inflammatory markers. Data were combined for the Rag1 -/- and Rag2 -/- animals as similar trends were seen in both strains.

### 2.4. Multiplexed Cytokine/Chemokine Analysis

The systemic levels of inflammatory markers (IL-17E/IL-25, GM-CSF, IFN-γ, MIP-3α/CCL20, IL-1β, IL-2, IL-4, IL-5, IL-6, IL-21, IL-22, IL28b, IL10, IL-23, IL-12p70, IL-27, IL-13, IL-15, IL-17A, IL-17F, IL33, IL-31, LT-α, TNF-α, CD40L) were assessed within the serum of animals according to manufacturer’s instructions using a commercially available Mouse TH17 Magnetic Bead Panel obtained from MILLIPLEX**^®^** (Millipore, Burlington, MA, USA). Briefly, 25 μL of serum sample was incubated overnight with 25 μL antibody-immobilized cytokine bead mix per well on a 96-well MAG-PLATE at 4 °C with shaking in the dark. Samples were washed twice with wash buffer and incubated at RT in the dark with detection antibodies for 1 h, followed by streptavidin-phycoerythrin for 30 min. Lastly, samples were washed twice and resuspended in 150 μL wash buffer prior to acquisition on the Luminex MAGPIX**^®^** instrument (MilliporeSigma, Burlington, MA, USA). Cytokine/chemokine expression levels were analyzed using the Milliplex^TM^ Analyst software version 5.1 (VigeneTech, Carlisle, MA, USA). For analysis and graphing purposes, the lower limit of quantification (LLOQ) value for a particular cytokine was used if the levels in the sample were measured to be <LLOQ.

### 2.5. Statistical Analysis

Data were analyzed using GraphPad Prism**^®^** version 9 (GraphPad Software, San Diego, CA, USA). Statistical significance of the difference between groups was calculated by unpaired parametric two-tailed t-test with the Welch’s correction, one-way or two-way analysis of variance (ANOVA) followed by post hoc analysis using Šidák’s or Dunnett’s (comparison across all groups) multiple comparison test as indicated in Figure legends. For all analyses, differences were considered to be not significant with *p* > 0.05. Significance was indicated in the graphs as follows: * *p* < 0.05, ** *p* < 0.01, *** *p* < 0.001 and ****: *p* < 0.0001.

## 3. Results

### 3.1. Clinical Symptoms in DSS-Induced Colitis Model

C57BL/6 mice (n = 10 per group) were given dextran sulphate sodium (DSS) within their drinking water at the indicated mass/volume percentage for a period of 7 days. Prior to identifying the blood-based inflammatory markers of disease, we wanted to confirm that the animals exhibited the typical symptoms of DSS-induced colitis. Indeed, animals exhibited clinical signs of disease over this period in a DSS dose-dependent manner. By Day 7, the mean Disease Activity Index (DAI) of each group was 1.3, 4.2, 5.5, 7.9 and 8.5 for untreated controls, 2%, 3%, 4% and 5% DSS-treated mice, respectively (Figure 1A). This measurement was defined by the cumulative scoring of weight loss, stool consistency and fecal blood occurrence parameters. The mean Weight Change (%) of each group on Day 7 was +2.70, +1.57, −5.22, −11.77 and −16.69 for untreated controls, 2%, 3%, 4% and 5% DSS-treated mice, respectively (Figure 1B). Given the semi-quantitative subjective manner of the assessment of stool consistency and fecal blood occurrence, the DAI scoring has an inherently higher standard deviation compared to the more quantitative weight loss parameter. After the conclusion of clinical monitoring on Day 7, the colons were isolated and measured for length prior to fixation for histopathological scoring. The mean Colon Length of each group decreased as expected in a dose-dependent manner: 6.08, 5.72, 4.98, 4.77 and 4.54 mm (Figure 1C). This matched up well with the mean Histopathological Score of each group: 0, 4.4, 8.1, 9 and 9.6 for untreated controls, 2%, 3%, 4% and 5% DSS-treated mice, respectively (Figure 1D).

### 3.2. Blood-Based Immune Protein Markers in DSS-Induced Colitis Model

Sera was collected from the mice on Day 7 just prior to euthanasia and analyzed for a panel of 25 inflammatory markers. A heat map (Figure 2A) indicating the relative Log2 increase above baseline values of each particular marker, show upregulated expression for a number of proteins in this panel upon DSS treatment. However, not all of these are of statistical significance. Looking more closely at the pg/mL concentrations within these animals, Figure 2B illustrates more clearly which systemically detectable inflammatory biomarkers were expressed differently within the DSS model. Despite some notable differences between the groups, IL-17E/IL-25, IFN-γ and IL-5 were not statistically different from control animals within any of the DSS-treated groups. The response was quite dose-dependent for IFN-γ levels, culminating in a mean level of 4.035 pg/mL within the 5% DSS group (*p* = 0.0816). Interestingly, IL-6 levels were nearly undetectable in the untreated or mildly colitic animals (2% DSS), whereas the groups receiving 3–5% DSS had >10-fold higher levels of IL-6 within their serum with means of 148–202 vs. 13 pg/mL for control animals (*p* ≤ 0.001). Similar trends were observed for IL-22, IL-13, LT-α and TNF-α, with small increases measured within the 2% DSS group and significantly higher systemic levels when the colitis worsens within the 3–5% DSS groups. The measured mean systemic concentrations of IL-22 were 122–180 in the 3–5% DSS-treated animals vs. 3 pg/mL in the control group (*p* < 0.05). In the 3%, 4% and 5% DSS groups, IL-13 concentration measured 122 (*p* = 0.1662), 139 (*p* < 0.01) and 127 (*p* = 0.0667) compared to 95 pg/mL within the control group; LT-α levels were 1.34 *×* 10^5^ (*p* = 0.1000), 1.49 *×* 10^5^ (*p* < 0.05) and 2.02 *×* 10^5^ (*p* < 0.01) compared to 0.45 *×* 10^5^ pg/mL within the control group; TNF-α levels were 4.6 (*p* = 0.1306), 5.1 (*p* < 0.05) and 5.1 (*p* < 0.05) compared to 3.3 pg/mL within the control group. Interestingly, the opposite trend was observed with CD40L, where serum in the control and 2% DSS groups had 82 and 78 pg/mL of protein, respectively, while 3–5% DSS groups all had lower levels with a mean < 66 pg/mL. Overall, this panel identified IL-6, IL-22, IL-13, LT-α and TNF-α as upregulated inflammatory markers and CD40L as a downregulated marker of DSS-induced experimental acute murine colitis.

### 3.3. Clinical Symptoms in T Cell Transfer Colitis Model

To induce experimental colitis, Rag1 -/- and Rag2 -/- mice (n = 5 of each per group) were injected i.p. with either a PBS vehicle solution or 5 × 10^5^ proinflammatory naïve T cells isolated from wild-type animals of the same genetic background (C57BL/6). Animals were monitored for clinical signs of disease for a period of 64 days. While the same DAI parameters as the above DSS colitis study were monitored, fecal scoring measurements (i.e., fecal consistency and blood occurrence) were largely negligible in both groups, with the feces looking generally normal over the course of the study (data not shown). The only DAI parameter that showed significant differences in this model was the weight monitoring. The mean Weight Change (%) of each group at endpoint was +22.10% and 0.34% for the control and T cell transfer animals, respectively (Figure 3A). Despite the clear trend, statistically significant weight loss was only achieved 8 weeks following T cell transfer. Overall, there was a high degree of variability in weight loss and severity of clinical disease in the animals receiving the proinflammatory T cells. After the 64 day period, mice were bled and colons isolated and measured, before being prepared for histopathological scoring. The colon lengths were comparable between the control animals (70.82 mm) and T cell transfer recipients (73.78 mm) (Figure 3B). More importantly for T cell transfer IBD models, the colon weight/length ratio has been shown to be a better proxy for disease severity [12]. The mean mass of colons from control animals was 0.20 g compared to the 0.40 g mass from T cell transfer mice (*p* = 0.0002) (Figure 3C). Accounting for the length of each colon, the mean Weight/Length Ratio for control animals was 2.82 *×* 10^−^^3^ g/mm compared to a mean of 5.54 *×* 10^−^^3^ g/mm from T cell transfer mice (*p* = 0.0004) (Figure 3D). The T cell transfer recipients also exhibited splenomegaly (Figure 3E), with control spleens weighing a mean of 2.09 *×* 10^−^^2^ g and spleens from T cell transfer animals weighing 7.16 *×* 10^−^^2^ g (*p* = 0.0003). The blinded Histopathological Score of control colons remained consistently 0, whereas the mean score of colons derived from T cell transfer recipients was 9.11 (*p* < 0.0001) out of a maximum score of 10 (Figure 3F). We observed fewer clinical signs of disease in the T cell transfer than the DSS model of IBD with only moderate amounts of body weight loss over 2 months; however, tissue analysis confirmed the inflammatory state of these mice. The variability and trends observed are consistent with previous studies [12,21].

### 3.4. Blood-Based Immune Protein Markers in T Cell Transfer Colitis Model

As before, sera from the euthanized mice were isolated and analyzed for the same panel of 25 inflammatory markers. A heat map (Figure 4A) indicates the relative Log2 increase above baseline values of each particular marker, showing upregulated trends for a variety of markers on this panel. Again, not all of these are of statistical significance. Looking closely at the pg/mL concentrations within these animals, there were 11 significantly upregulated inflammatory cytokines/chemokines when compared to control animals (Figure 4B). These include IFN-γ (T cell transfer vs. Untreated; mean of 147 vs. 7.17 pg/mL; *p* < 0.05), MIP-3α/CCL20 (mean of 2601 vs. 76.11 pg/mL; *p* < 0.05), IL-2 (mean of 31 vs. 10 pg/mL; *p* < 0.0001), IL-4 (mean of 1.4 vs. 0.9 pg/mL; *p* < 0.05), IL-5 (mean of 250 vs. 94 pg/mL; *p* < 0.05), IL-6 (mean of 279 vs. 28 pg/mL; *p* < 0.01), IL-22 (mean of 830 vs. 60 pg/mL; *p* < 0.001), IL-10 (mean of 26 vs. 15 pg/mL; *p* < 0.05), IL-13 (mean of 447 vs. 136 pg/mL; *p* < 0.001), IL-17A (mean of 119 vs. 73 pg/mL; *p* < 0.01) and TNF-α (mean of 200 vs. 10 pg/mL; *p* < 0.01). In addition, there were 2 biomarkers of interest with nearly significant (*p* < 0.10) alterations during the disease state: IL-31 (mean of 104 vs. 67 pg/mL; *p* = 0.0517) and IL-27 (mean of 2129 vs. 1542 pg/mL; *p* = 0.0754). Given their significance in the DSS model and the similar trends across both Heatmaps, we also took a closer look at LT-α and CD40L (Figure 2A and Figure 4A): LT-α (5.52 *×* 10^5^ vs. 4.67 *×* 10^5^; *p* = 0.3819) and CD40L (151 vs. 221 pg/mL; *p* = 0.1810). The trends with these biomarkers are similar to what was seen in the DSS model, but without statistical significance. Overall, this panel has identified IFN-γ, MIP-3α/CCL20, IL-2, IL-4, IL-5, IL-6, IL-22, IL-10, IL-13, IL-17A and TNF-α as significantly upregulated inflammatory markers within the T cell transfer murine model of experimental IBD. 

One of the main advantages of this type of blood-based analysis is to confidently monitor disease state during the course of a study without needing to euthanize some or all of the animals. As an example, we had analyzed sera collected at Day 35 of the study above, as the body weights of these mice began to diverge, for the same 25 inflammatory markers. Similarly to endpoint, there were a number of statistically significant differences in the cytokine/chemokine profiles within the blood of these animals (Figure 5). Altogether, there were 8 significantly upregulated inflammatory proteins when compared to control animals. These include IFN-γ (T cell transfer vs. Untreated; mean of 333 vs. 6 pg/mL; *p* < 0.01), IL-2 (mean of 27 vs. 7 pg/mL; *p* < 0.01), IL-4 (mean of 3.8 vs. 1.6 pg/mL; *p* < 0.05), IL-5 (mean of 185 vs. 45 pg/mL; *p* < 0.01), IL-6 (mean of 336 vs. 19 pg/mL; *p* < 0.01), IL-22 (mean of 947 vs. 41 pg/mL; *p* < 0.01), IL-10 (mean of 32 vs. 10 pg/mL; *p* < 0.01) and TNF-α (mean of 251 vs. 7 pg/mL; *p* < 0.001). In addition, there were 3 biomarkers of interest with nearly significant (*p* ≤ 0.10) alterations during the disease state: MIP-3α/CCL20 (mean of 510 vs. 41 pg/mL; *p* = 0.0521), GM-CSF (mean of 56 vs. 29 pg/mL; *p* = 0.0603), and IL-13 (274 vs. 155 pg/mL; *p* = 0.1005). No differences were observed with LT-α (6.88 *×* 10^5^ vs. 6.36 *×* 10^5^; *p* = 0.7775) and CD40L (129 vs. 118 pg/mL; *p* = 0.7755; data not shown). It is interesting to note some different trends at this earlier stage of disease when compared to the endpoint: mainly with the upregulation of GM-CSF at Day 35, and levels of IL-31 or IL-27. 

## 4. Discussion

Due to the complex multifaceted nature of inflammatory bowel disease (IBD), a number of animal models have been developed and employed with the aim of recapitulating different aspects of the human disease state [10]. As there are phenotypic differences between the models, it is not surprising that some therapies show efficacy within certain animal models but fail within others—highlighting the value of multiple models of disease [22]. There is a great deal of effort in preclinical testing of novel therapeutics in the research community [4]. Unfortunately, the measurement of outward clinical signs of disease are not sufficient to fully capture severity of disease in certain IBD models. In addition, as colitis can be microbiota-dependent, disease severity may vary across institutions [12,21]. The gold standard remains post-mortem analyses, such as histopathological scoring and colonic physical characteristics. As the basis of these disease symptoms begins at the molecular/cellular levels, it can be greatly informative to monitor signs of disease at this level. For example, inflammatory markers are especially useful within models such as the T cell transfer IBD model wherein the clinical signs are inherently more subtle and variable [12,19,21]. The use of dexamethasone within T cell transfer colitis is not associated with a significant improvement in body weight, but can alleviate induction of TNF-α and upregulate IL-17A when compared to untreated animals [19]. Side effects of common therapeutics, such as corticosteroids, include modulation of appetite and weight outcomes, which further highlights the challenges with only using clinical disease monitoring as a readout for treatment efficacy [23]. 

With the availability of sensitive assays and the ability to use easily accessible blood samples, we demonstrated that the measurement of serum cytokines/chemokines could be a powerful tool to measure disease severity in two popular models of IBD, acute disease mediated by DSS-induced innate immunity and chronic disease mediated by proinflammatory T cell transfer. Combined with ongoing monitoring of disease activity and post-mortem analyses, we have illustrated several key differences between these two models of disease (Table 3). For example, acute DSS-induced colitis can be effectively monitored using a Disease Activity Index (DAI) Score which includes weight loss, fecal consistency and fecal blood occurrence (Figure 1A), whereas T-cell-transfer-mediated chronic colitis induces a more variable weight loss and negligible differences in fecal phenotypes. Both models of experimental IBD did exhibit the expected effects on colon size and histopathological inflammation as previous studies [11,12]. Furthermore, by including both female and male mice, we were able to confirm the finding that female mice exhibit a slightly milder form of disease as observed by Histopathological Scoring (Figure 1D) [24]. There was a clear dose-dependent trend in the severity of clinical signs observed within the acute DSS model (Figure 1); however, there were no statistically significant differences in the levels of induced cytokines when comparing within animals treated with 3–5% DSS. Whether this is indicative of the localized nature of disease in this preclinical model, which is similar to ulcerative colitis, or specific to the mucosal layers impacted by DSS treatment is not yet known. Investigations into the systemic levels of inflammatory markers within DSS- or TNBS-induced colitis, which exhibits more of a Crohn’s Disease-like phenotype and a more broad transmural inflammation, have shown slight differences in their inflammation profiles [18,25,26]. On the other hand, for T-cell-transfer-mediated colitis, Rag1 -/- and Rag2 -/- are often used interchangeably due to their phenotypic similarities. Herein we compared these genotypes side-by-side. While there may be some minor differences, the overall trends in disease-related readouts were the same between both strains (Figure 3). A larger number of animals would be required to confirm whether there are any statistical differences. In both models of experimental colitis, there were no obvious differences between either sexes (DSS) or genotypes (T cell transfer) tested when it comes to systemically quantified biomarkers of inflammation (Figure 2, Figure 4 and Figure 5).

For DSS-induced acute colitis, the serological screen for inflammatory markers identified IL-6, IL-22, IL-13, LT-α, TNF-α and CD40L. With a larger degree of fold-induction in treated vs. control animals, our data indicate that IL-6, IL-22 and LT-α would be the most useful markers in a DSS-induced colitis model. Given the innate nature of this disease model, these cytokines are likely produced by monocytes, macrophages, mast cells, innate lymphoid cells, natural killer cells and intestinal epithelial cells [28]. One cannot exclude the activity of lymphocytes as seen in low dose DSS treatment [27], though the short timeline for this acute DSS treatment may limit their involvement. While not statistically significant in this study, both IFN-γ and IL-17A were upregulated in a dose-dependent manner (Figure 2). The top systemically detectable cytokines within the T cell transfer model of disease align with a multipronged T cell response; Th1 (IFN-γ, TNF-α, IL-2, IL-10, GM-CSF), Th2 (IL-4, IL-5, IL-6 IL-13), Th17 (IL-17A, IL-22, MIP3α/CCL20) and Th22 (TNF-α, IL-13, IL-22) [29,30]. Although one cannot rule out an activated innate immune system, given the interplay and potential other sources of many of these cytokines. For example, GM-CSF activated monocytes have been shown to interact with T cells in the context of colitis [31]. It is noteworthy that the adoptive T cell transfer model not only had more statistically relevant inflammatory markers, but also higher concentrations of all markers tested (Figure 4B and Figure 5). This is especially noticeable in IL-13, IL-22, TNF-α and IFN-γ where the levels detected within the serum were 2–80 fold higher in the chronic T cell transfer model compared to the acute DSS model. This was true at both time points tested for the T cell transfer model. There are a number of differing factors that may contribute to this, such as the more localized nature of inflammation within the DSS-treated mice, the predominant adaptive vs. innate immune cell involvement and/or the acute vs. chronic states of these animals. 

Overall, our results above which demonstrate an induction of TNF-α, IFN-γ, IL-17A and IL-6 align well with some previously published results investigating the levels of systemic biomarkers within preclinical mouse models [18,19]. Indeed, a handful of these markers are typically evaluated in a targeted fashion (i.e., ELISA or qRT-PCR) for the qualification of treatments in these preclinical models, such as fecal microbiota transplantation or novel therapeutics [14,15,16,17]. It also highlighted changes in other markers that are not often monitored in preclinical studies (i.e., MIP3α/CCL20, LT-α, CD40L, IL-31, IL-27, GM-CSF, etc.), but that do have biological relevance in human IBD [32,33,34,35,36]. In the cases of IL-27 and IL-31, there is evidence of upregulation within human studies and cytokine/receptor knockout models, but they have not been previously described within these animal models as part of a systemic analysis [37,38,39]. LT-α has been recently shown to be a contributor to TNF-α-independent intestinal damage in preclinical models, potentially explaining the proportion of non-responders to anti-TNF-α therapy amongst human IBD patients [40]. GM-CSF is highly important in the microbial cross-talk observed in human IBD [41,42,43] and was identified as an upregulated marker in the T cell transfer colitis model above (Figure 5). With this in mind, it could be interesting to look more closely at the kinetics of GM-CSF and other inflammatory markers in long-term chronic models, such as T cell transfer, and correlate them with disease progression. Further studies are warranted to validate the presence of the markers identified herein within other commonly used preclinical models of inflammatory bowel disease.

## 5. Conclusions

Altogether, this study’s evaluation of a panel of 25 immune proteins in the blood from two commonly used preclinical models of inflammatory bowel disease has improved our understanding of the systemically circulating immune biomarkers that can be induced and how they compare between the models. With the unique comparative nature of this analysis, it promises to be a useful tool to help facilitate the rational analyte design when looking to evaluate therapeutics in particular preclinical IBD models. Furthermore, with the use of more easily translatable methods such as blood-based biomarker analysis in preclinical and clinical studies, it will be possible to better link the activity and mechanisms of action of various drugs in these different contexts. 

## Figures and Tables

**Figure 1 biomedicines-11-00140-f001:**
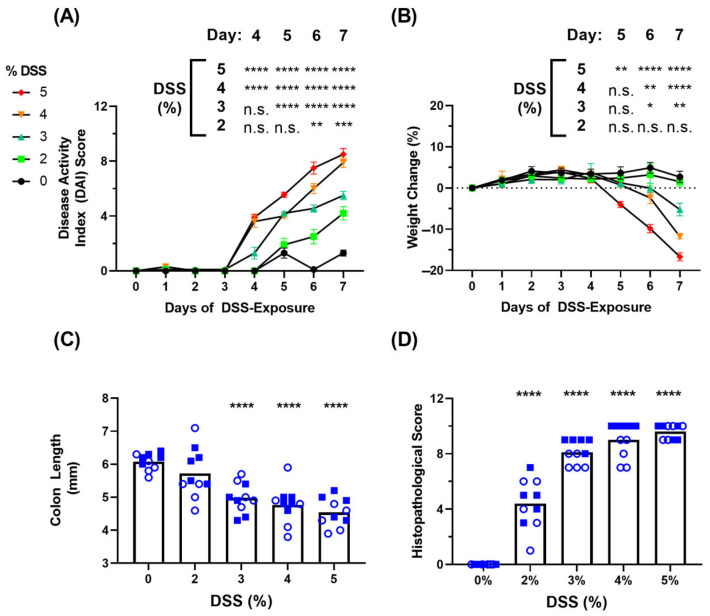
Clinical signs of disease within the acute, DSS-mediated experimental colitis model. C57BL/6 mice (n = 10/group) were exposed to dextran sulphate sodium (DSS) within drinking water (0–5% *m*/*v*) for a period of 7 days. The Disease Activity Index (DAI) (Panel (**A**)) and Body Weight Change (%) (Panel (**B**)) for each of the groups during the course of DSS treatment is shown. At the end of the study period, the colons were examined for their length (Panel (**C**)) before being fixed, H&E stained and analyzed for a Histopathological Score (Panel (**D**)). Grouped data is presented as mean + standard error of mean (SEM). In panels (**C**,**D**), empty circles and filled squares represent female and male mice, respectively. Statistical significance of differences are compared relative to the control (0% DSS) group and are shown as *: *p* < 0.05, **: *p* < 0.01, ***: *p* < 0.001, ****: *p* < 0.0001 by two-way ANOVA followed by Šidák’s (Panel (**A**)) or Dunnett’s (Panel (**B**)) multiple comparisons test or by one-way ANOVA followed by Dunnett’s multiple comparison test (Panels (**C**,**D**)).

**Figure 2 biomedicines-11-00140-f002:**
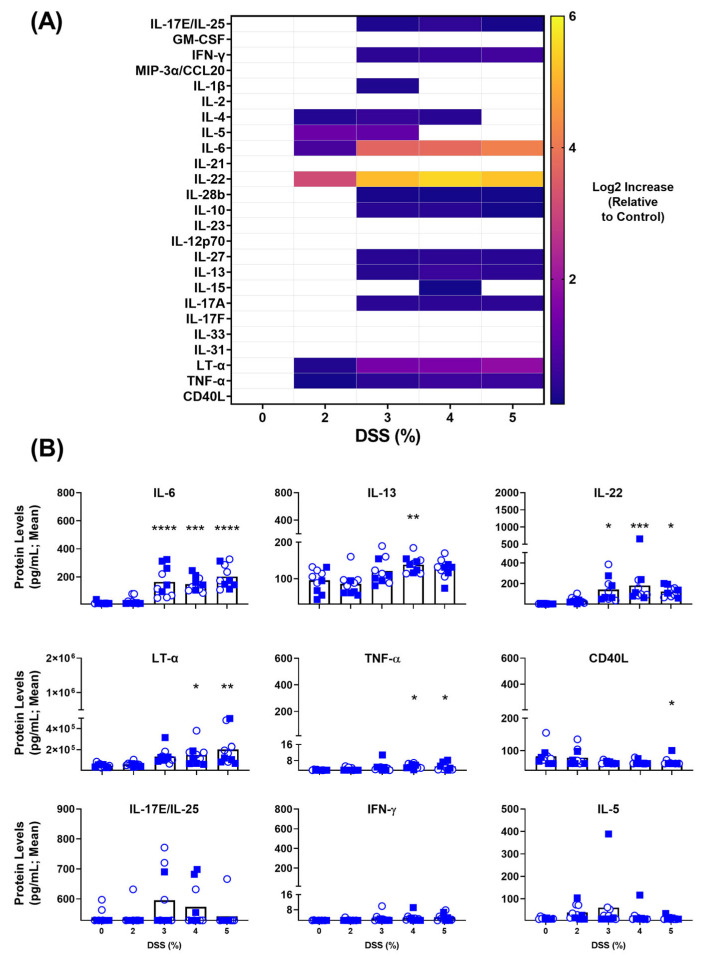
Inflammatory markers dysregulated within acute, DSS-mediated colitis model. C57BL/6 mice (n = 10/group) were exposed to dextran sulphate sodium (DSS) within drinking water (0–5% *m*/*v*) for a period of 7 days. At the end of the study period, serum was collected and used for a multiplexed cytokine/chemokine analysis. Raw concentration values were determined through use of a standard curve using the Milliplex™ Analyst software. These values were normalized to the control animals for a baseline, and the mean Log2 Increase was plotted (**A**). Grouped raw concentration data (**B**) are presented as mean + standard error of mean (SEM). Circles (empty) and squares (filled) represent female and male mice, respectively. Statistical significance of differences are compared relative to the control (0% DSS) group and are shown as *: *p* < 0.05, **: *p* < 0.01, ***: *p* < 0.001 & ****: *p* < 0.0001 by one-way ANOVA followed by Dunnett’s multiple comparison test. The y-axis is set to start at the LLOQ of the particular cytokine in our assay.

**Figure 3 biomedicines-11-00140-f003:**
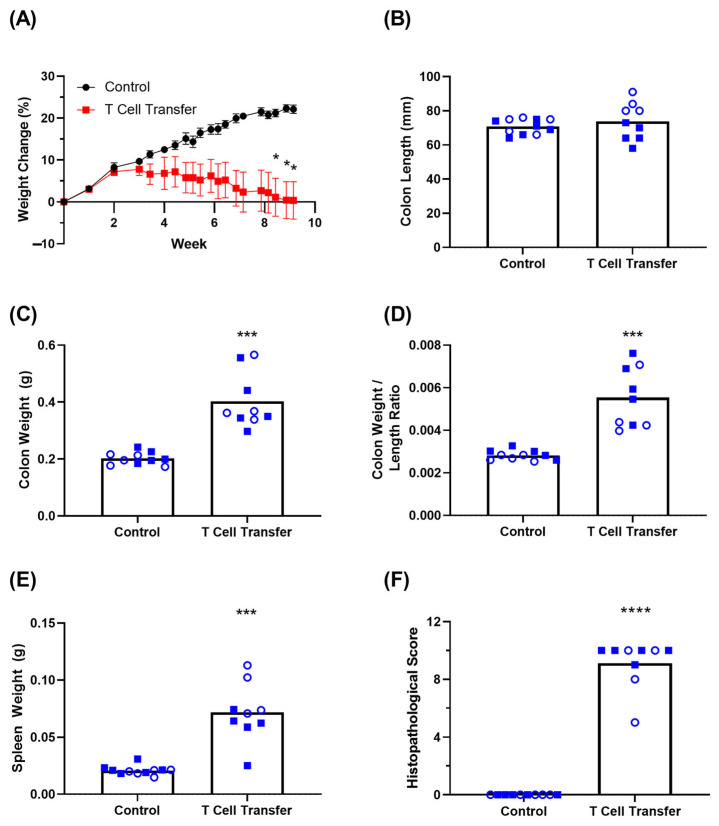
Clinical signs of disease within the chronic, adoptive T cell transfer experimental colitis model. Rag KO mice (n = 10/group) were injected i.p. with proinflammatory (CD4+ CD25^LOW^CD45RB^HIGH^ isolated from spleens of C57BL/6 mice. Body Weight Change (%) over the course of the study is shown (Panel (**A**)). For animals euthanized during the course of the study, the last recorded weight loss was used for time points post-euthanasia. Colons were collected once animal were euthanized at the end of the study period or for reaching a humane endpoint, and examined for their length (Panel (**B**)), weight (Panel (**C**)) and related to one another for a Weight/Length Ratio (Panel (**D**)). Isolated spleens were also weighed (Panel (**E**)). Colons were then fixed, H&E-stained and analyzed for a Histopathological Score (Panel (**F**)). Data are presented as mean + standard error of mean (SEM). Circles and squares represent Rag1 -/- and Rag2 -/- mice, respectively. Statistical significance of differences is compared relative to the control group and is shown as *: *p* < 0.05, ***: *p* < 0.001, ****: *p* < 0.0001 by two-tailed unpaired parametric *t*-tests with the Welch’s correction.

**Figure 4 biomedicines-11-00140-f004:**
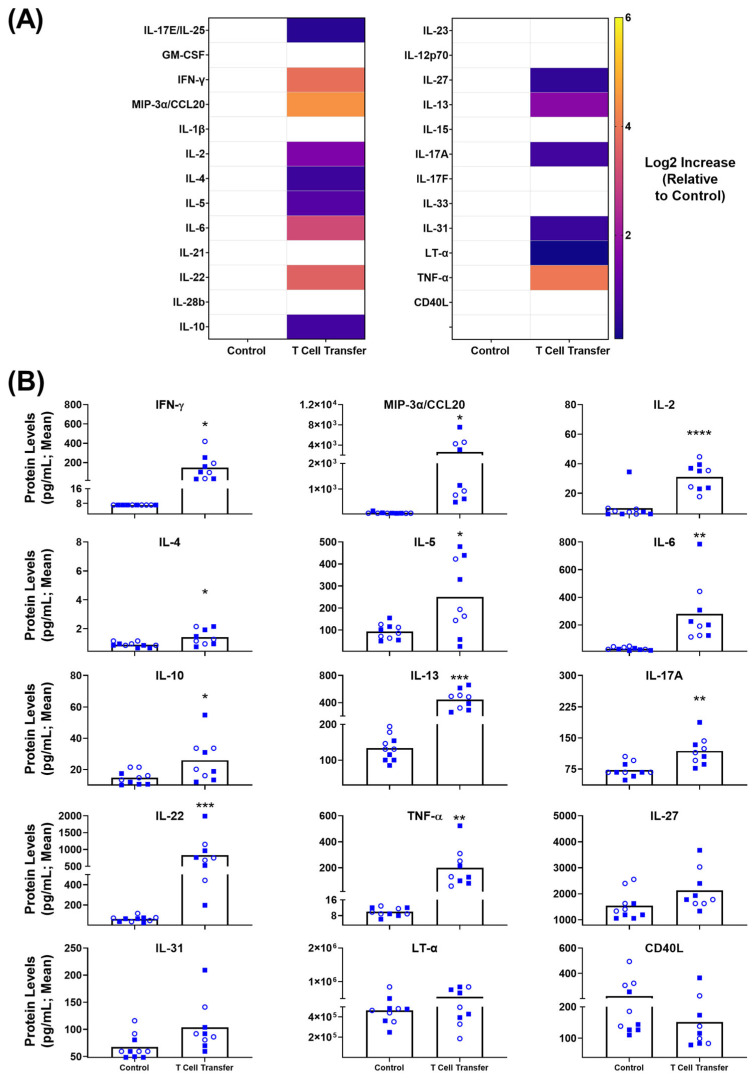
Inflammatory markers dysregulated within chronic, T cell transfer colitis model. Rag KO mice (n = 10/group) were injected i.p. with proinflammatory (CD4+ CD25^LOW^CD45RB^HIGH^ isolated from spleens of C57BL/6 mice. At the time of euthanasia (Day 64 or humane endpoint), serum was collected and used for a multiplexed cytokine/chemokine analysis. Raw concentration values were determined through use of a standard curve using the Milliplex™ Analyst software. These values were normalized to the control animals for a baseline, and the mean Log2 Increase (**A**) was plotted. Grouped data of raw concentrations (**B**) are presented as mean + standard error of mean (SEM). Circles (empty) and squares (filled) represent Rag1 -/- and Rag2 -/- mice, respectively. Statistical significance of differences are compared relative to the control group and are shown as *: *p* < 0.05, **: *p* < 0.01, ***: *p* < 0.001 & ****: *p* < 0.0001 by two-tailed unpaired parametric *t*-tests with the Welch’s correction.

**Figure 5 biomedicines-11-00140-f005:**
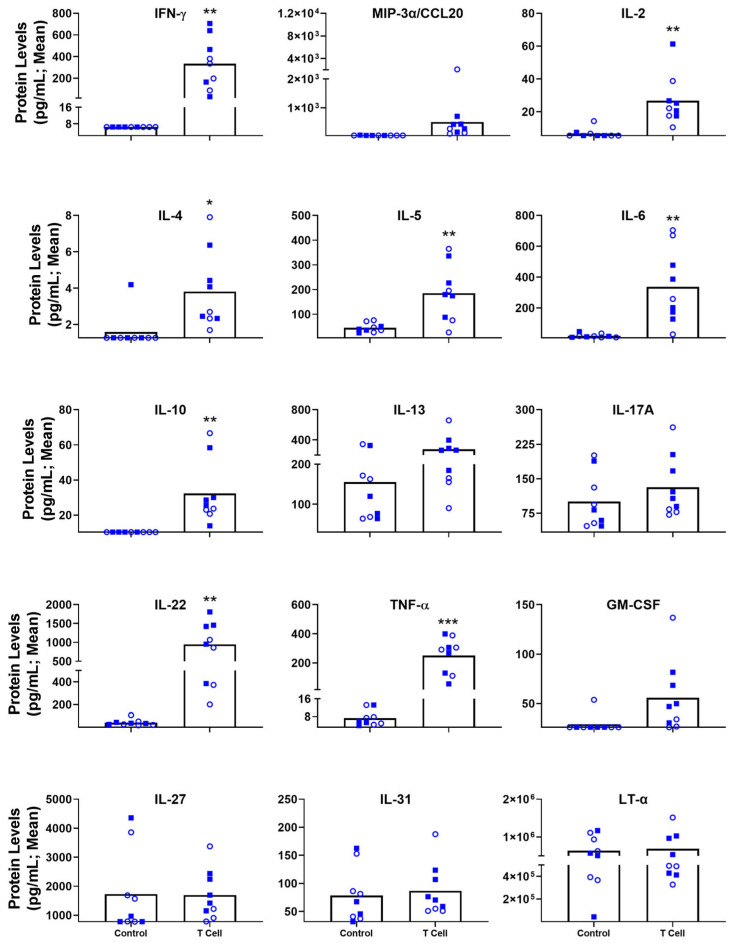
Individual inflammatory markers dysregulated within chronic, T cell transfer colitis model at Day 35 post-T cell transfer. Rag KO mice (n = 10/group) were injected i.p. with proinflammatory (CD4+ CD25^LOW^CD45RB^HIGH^ isolated from spleens of C57BL/6 mice. Serum was collected on Day 35 and used for a multiplexed cytokine/chemokine analysis. Raw concentration values were determined through use of a standard curve using the Milliplex^TM^ Analyst software and plotted. Grouped data is presented as mean + standard error of mean (SEM). Circles (empty) and squares (filled) represent Rag1 -/- and Rag2 -/- mice, respectively. Statistical significance of differences is compared relative to the control group and is shown as *: *p* < 0.05, **: *p* < 0.01 & ***: *p* < 0.001 by two-tailed unpaired parametric *t*-tests with the Welch’s correction.

**Table 1 biomedicines-11-00140-t001:** Disease activity index (DAI) scoring metrics.

	Weight Loss (%)	Stool Consistency	Fecal Blood Occurrence
0	<0	Normal	No blood on feces
1	1–5	Soft, but still formed	Slight traces of blood on feces
2	5–10	Very soft	Pronounced traces of blood on feces
3	10–20	N/A	N/A
4	>20	Diarrhea	Gross rectal bleeding

N/A: Not applicable.

**Table 2 biomedicines-11-00140-t002:** Histopathological scoring metrics.

	Inflammatory Cell Infiltrate	Intestinal Architecture	Extent of Organ Inflammation (%)
0	Normal	Normal	None
1	Mild/Mucosal	Focal erosion	0–25
2	Moderate/Mucosal and submucosal	Erosions/focal ulcerations	25–50
3	Marked/Transmural	Extended ulcerations/ granulation/ pseudopolyps	50–75
4	N/A	N/A	75–100

N/A: Not applicable.

**Table 3 biomedicines-11-00140-t003:** Phenotypes of acute (DSS-mediated) and chronic (T-cell-transfer-mediated) IBD models.

	Acute (DSS)	Chronic (T Cell Transfer)
Timeline	Days	Weeks
Immunity	Innate ^i^	Adaptive
Disease Activity Index (DAI)	✓	✗
Weight Loss *	✓	✓
Fecal Consistency	✓	✗
Fecal Blood	✓	✗
Colon size **	✓	✓
Histopathological Scoring	✓	✓
Systemically Elevated Immune Markers	IL-6, IL-22, IL-13, LT-α, TNF-α (IFN-γ)	IFN-γ, MIP-3α/CCL20, IL-2, IL-4, IL-5, IL-6, IL-22, IL-10, IL-13, IL-17A, TNF-α (IL-31, IL-27, GM-CSF ^t^)
Systemically Downregulated Immune Marker	CD40L	

* Weight loss is present, but more variable within the T cell transfer model. ** Colon shrinkage in mass and length is expected within the DSS model, whereas increased colon mass-to-length ratio is expected within the T cell transfer model due to the granulomatic nature of the inflammation. ^i^ The acute DSS model used herein has been shown to rely mostly on the innate immune response; however, there may be some interplay of adaptive immunity within immunocompetent mice. Induction of IBD symptoms by low dose DSS treatment over a longer time period than within this report has been shown to rely more substantially on the adaptive immune response [27]. ^t^ GM-CSF seems to be an early indicator of disease progression, but is near baseline levels at endpoint. () Parentheses signify trending, but not statistically significant proteins (0.05 < *p* < 0.10).

## Data Availability

The data presented in this study are available on request from the corresponding author.

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
