# Peer review of "Blood-Based Immune Protein Markers of Disease Progression in Murine Models of Acute and Chronic Inflammatory Bowel Disease"

_biomedicines, 2023, doi:10.3390/biomedicines11010140_

Round 1

Reviewer 1 Report

In this study, the authors analyze the expression profile of a panel of cytokines and chemokines in blood in the most used colitis mouse models, DSS and T cell transfer. Although the approach to finding biomarkers for diagnosis and prognosis of IBD in the blood is very interesting, this study does not support an important impact in the field since these types of analyses are amply carried out by other groups as the authors confirmed in the references included.

It is highlighted that this study contributes to the field with the analysis of levels of cytokines and chemokines in different ranges of DSS. This is interesting since this model could mimic UC or CD depending on the mucus layer affected. Histology analysis is necessary to compare cytokine/chemokine levels as well as a rational discussion with the levels of these proteins in human IBD blood.

Additionally, they measured cytokine/chemokine levels at different times in the T cell transfer model but they do not explain the results in detail and they do not compare them with human IBD.

On other hand, the comparison between the two models would be a good point in this manuscript since they are models that mimic different features and immune responses of human IBD. They could make figures with cytokine/chemokine analysis from both colitis models.

Remarkably, the comparison of the protein levels between male and female and Rag1 and Rag2 knock out mice in this type of analysis in the T cell transfer model.

The discussion section should be intensely improved since they did not compare the levels of proteins between the models, the range of DSS, and the different moments in T cell transfer as well as with human IBD

The conclusion section must be improved

Minor points

In multiplexed cytokine/chemokine analysis section in Materials and Methods, they mention that they isolate TH17 cells. Is that right? Thus, Have all the results been obtained from TH17 cells? They should mention it throughout the text. If yes, some conclusions are wrong.

They should join heatmaps and protein level assays in the same figures to make the comparisons easier for the reader

Fecal consistency and blood are usually published that are affected in the T cell transfer model

As it is also amply published, not only innate immune responses are implicated in the DSS model as mentioned in table 3.

Why did they change DSS every 2 days? They should mention it.

How did they analyze traces of blood in feces?

They should mention in the result section the day that they carried out the cytokine/chemokine analysis in the DSS model.

When they mention “wild type mice” in some parts of the manuscript they should mention if they are C57BL6 or RAG mice.

They should use similar scale sizes in the figures  

References should be more recent

Author Response

Response to Reviewer 1 Comments:

Comments and Suggestions for Authors

In this study, the authors analyze the expression profile of a panel of cytokines and chemokines in blood in the most used colitis mouse models, DSS and T cell transfer. Although the approach to finding biomarkers for diagnosis and prognosis of IBD in the blood is very interesting, this study does not support an important impact in the field since these types of analyses are amply carried out by other groups as the authors confirmed in the references included.

We thank the reviewer for their many helpful suggestions and comments.  While previous studies have identified some of the markers in our report or have used knockout models to demonstrate the importance of certain cytokines in developing IBD, we feel that our work does reveal new insights.  The novelty in this study lies mainly in the direct comparison of blood-based biomarkers of disease between two commonly used IBD preclinical models.  Also, to our knowledge, this is the first demonstration of certain serological markers such as GM-CSF, LT-α, CD40L, IL-27 and IL-31, which have been implicated in human IBD, being dysregulated specifically in these models. Many of these cytokines/chemokines within this panel are typically excluded from targeted panel design, therefore this report provides impact in the area of rational panel design for future studies of these and similar preclinical mouse models.

It is highlighted that this study contributes to the field with the analysis of levels of cytokines and chemokines in different ranges of DSS. This is interesting since this model could mimic UC or CD depending on the mucus layer affected. Histology analysis is necessary to compare cytokine/chemokine levels as well as a rational discussion with the levels of these proteins in human IBD blood.

Typically, whether it is acute or chronic (intermittent low-dose) DSS colitis, it is accepted in the field that this type of chemical treatment induces a more ulcerative colitis (UC) phenotype. Whereas treatment with TNBS is more representative of Crohn’s disease (CD). This is an interesting discussion point and has been added into the manuscript as follows:

“Whether this is indicative of the localized nature of disease in this preclinical model, which is similar to ulcerative colitis, or specific to the mucosal layers impacted by DSS treatment is not yet known[24, 25]. Investigations into the systemic levels of inflammatory markers within DSS- or TNBS-induced colitis, which exhibits more of a Crohn’s Disease-like phenotype and a more broad transmural inflammation, have shown slight differences in their in-flammation profiles[18, 24, 25].”

Additionally, they measured cytokine/chemokine levels at different times in the T cell transfer model but they do not explain the results in detail and they do not compare them with human IBD.

On other hand, the comparison between the two models would be a good point in this manuscript since they are models that mimic different features and immune responses of human IBD. They could make figures with cytokine/chemokine analysis from both colitis models.

While the focus of our manuscript is directed towards the study of these two particular preclinical mouse models, and not the human disease itself - the reviewer makes a good point to provide more linkages to the human state of disease. The discussion has been modified to include more links between the inflammatory markers detected and human IBD. See below for the exact text within the manuscript that has been added:

“In the cases of IL-27 & IL-31, there is evidence of upregulation within human studies and cytokine/receptor knockout models, but they have not been previously described within these animal models as part of a systemic analysis [36–38]. LT-α has been recently shown to be a contributor to TNF-α-independent intestinal damage in preclinical models, potentially explaining the proportion of non-responders to anti-TNF-α therapy amongst human IBD patients[39]. GM-CSF is highly important in the microbial cross-talk observed in human IBD[40–42] and was identified as an upregulated marker in the T cell transfer colitis model above (Figure 7). With this in mind, it could be interesting to look more closely at the kinetics of GM-CSF and other inflammatory markers in long-term chronic models, such as T cell transfer, and correlate them with disease progression.”

The discussion section should be intensely improved since they did not compare the levels of proteins between the models, the range of DSS, and the different moments in T cell transfer as well as with human IBD

This is a good point, while we alluded to this we did not directly compare magnitudes of responses in the original submission. See below for the text within the discussion where this is directly commented on:

“It is noteworthy that the adoptive T cell transfer model not only had more statistically relevant inflammatory markers, but also higher concentrations of all markers tested (Figure 6 & 7). This is especially noticeable in IL-13, IL-22, TNF-α and IFN-γ where the levels detected within the serum were 2-80 fold higher in the chronic T cell transfer model compared to the acute DSS model. This was true at both time points tested for the T cell transfer model.”

The conclusion section must be improved

The conclusion has been reworked in an effort to improve its clarity and better summarize the study’s findings and impact.

Minor points

In multiplexed cytokine/chemokine analysis section in Materials and Methods, they mention that they isolate TH17 cells. Is that right? Thus, Have all the results been obtained from TH17 cells? They should mention it throughout the text. If yes, some conclusions are wrong.

The commercially available kit is named the “Mouse TH17 Magnetic Bead Panel”, whereby magnetic beads are coated with antibodies to analyze cytokines / chemokines within a biological sample that are associated with Th17 cells. No isolation of TH17 cells was performed.

They should join heatmaps and protein level assays in the same figures to make the comparisons easier for the reader

While we appreciate this perspective given the direct link between these figures, we believe the differences in scale between the heatmaps (Figures 2 & 5) and the individual cytokine graphs (Figures 3, 6 & 7) are too drastic to merge into a single figure. This will cause some distortions and will be visually unappealing

Fecal consistency and blood are usually published that are affected in the T cell transfer model

Indeed, if we did observe any significant metrics within these parameters it would have been plotted and included in Figure 4. However, within the T cell transfer model, we observed almost no impact to fecal consistency and blood scoring. Though this may be explained by the differences seen between institutional gut microbiota in the animal colonies as discussed in the manuscript. We did discuss our findings with regards to lack of changes in fecal consistency and blood in the Results:

“While the same DAI parameters as the above DSS colitis study were monitored, fecal scoring measurements (i.e. fecal consistency and blood occurrence) were largely negligible in both groups, with the feces looking generally normal over the course of the study (data not shown).”

These findings highlight the main focus of the manuscript, the need to monitor more evident indicators of disease. 

As it is also amply published, not only innate immune responses are implicated in the DSS model as mentioned in table 3.

This is a good point, we should not exclude the possibility of lymphocyte contributions within the immunocompetent animals. However, given the timeline of the study, we believe the data in the literature support a limited contribution by the adaptive immune response. See the text below for our modifications to the manuscript.

Table 1 includes this footnote:

iThe acute DSS model used herein has been shown to rely mostly on the innate immune response, however there may be some interplay of adaptive immunity within immunocompetent mice. Induction of IBD symptoms by low dose DSS treatment over a longer time period than within this report has been shown to rely more substantially on the adaptive immune response[26].”

The discussion includes this statement:

“One cannot exclude the activity of lymphocytes as seen in low dose DSS treatment[26], though the short timeline for this acute DSS treatment may limit their involvement.”

See reference: Kim, Tae Woon, Jae Nam Seo, Young Ho Suh, Hyo Jin Park, Ju Hyun Kim, Ji Young Kim, and Kwon Ik Oh. 2006. Involvement of lymphocytes in dextran sulfate sodium-induced experimental colitis. World Journal of Gastroenterology 12: 302–305. https://doi.org/10.3748/wjg.v12.i2.302.

Why did they change DSS every 2 days? They should mention it.

            This has been clarified in the text as follows:

“To minimize concerns over potential instability of DSS while in solution over time, drinking water solutions were replaced at 2 day intervals for the duration of the experiment.”

How did they analyze traces of blood in feces?

            This was done visually and was clarified in the text.

            “Mice and feces were visually monitored daily for clinical signs of disease…”

They should mention in the result section the day that they carried out the cytokine/chemokine analysis in the DSS model.

            This was clarified in the Results section as suggested.

“Sera was collected from the mice on Day 7 just prior to euthanasia and analyzed for a panel of 25 inflammatory markers.”

When they mention “wild type mice” in some parts of the manuscript they should mention if they are C57BL6 or RAG mice.

            This was corrected throughout the manuscript.

They should use similar scale sizes in the figures.

            Figures 3, 6 & 7 were all adjusted to maintain the same scale size for the graphs.

References should be more recent

More recent references were added. Roughly half of the citations are now from the last 5 years.

Reviewer 2 Report

The authors have evaluated the systemic inflammatory profiles of two commonly employed murine models of inflammatory bowel disease: DSS-mediated acute colitis and chronic colitis induced by T cell transfer. 

Inflammatory bowel disease (IBD) is a chronic ailment affecting millions of people worldwide. This study adds promising cues to better understand the prognostic and diagnostic markers in case of IBD. The rationale behind the study as mentioned is the selection of serum-based markers for the evaluation of therapeutics in IBD models and highlights key biological differences between these two commonly used mouse models of IBD.Overall the design and scientific content of the study is robust. 

The manuscript is well written and the data is well presented as well. I would like to add however, the novelty of the study is bit lacking as this field of inflammatory markers/ immune response markers is very well studied. But authors done a good job in comparing two types of murine IBD models and presenting the conclusions drawn from each. Since it is commonly studied topic, please add a few more relevant citations.

Author Response

Response to Reviewer 2 Comments:

Comments and Suggestions for Authors

The authors have evaluated the systemic inflammatory profiles of two commonly employed murine models of inflammatory bowel disease: DSS-mediated acute colitis and chronic colitis induced by T cell transfer. 

Inflammatory bowel disease (IBD) is a chronic ailment affecting millions of people worldwide. This study adds promising cues to better understand the prognostic and diagnostic markers in case of IBD. The rationale behind the study as mentioned is the selection of serum-based markers for the evaluation of therapeutics in IBD models and highlights key biological differences between these two commonly used mouse models of IBD. Overall the design and scientific content of the study is robust. 

The manuscript is well written and the data is well presented as well. I would like to add however, the novelty of the study is bit lacking as this field of inflammatory markers/ immune response markers is very well studied. But authors done a good job in comparing two types of murine IBD models and presenting the conclusions drawn from each. Since it is commonly studied topic, please add a few more relevant citations.

We thank the Reviewer for your insight and comments, of which we are mostly on the same page. While previous studies have identified some of the markers in our report or have used knockout models to demonstrate the importance of certain cytokines in developing IBD, we feel that our work does reveal new insights.  The novelty in this study lies mainly in the direct comparison of blood-based biomarkers of disease between two commonly used IBD preclinical models.  Also, to our knowledge, this is the first demonstration of certain serological markers such as GM-CSF, LT-α, CD40L, IL-27 and IL-31, which have been implicated in human IBD, being dysregulated specifically in these models. Many of these cytokines/chemokines within this panel are typically excluded from targeted panel design, therefore this report provides impact in the area of rational panel design for future studies of these and similar preclinical mouse models.

In our revisions and improvement of the discussion particularly, more recent references were added. Roughly half of the citations are now from the last 5 years.

Round 2

Reviewer 1 Report

Article has improved considerably. Authors have elaborated more focused discussion and conclusion sections together with the updating of the references

 Minor points:

1.     I insist on the joining of heatmap of Figure 2 with the cytokine graphs in Figure 3 in one Figure. By other side, I also insist on the joining of the heatmap on Figure 5 with the cytokine graphs in Figure 6 in the final version of the manuscript to make easier for the reader

2.     I insist on the same scale sizes in the figure to be able to compare the cytokine levels between the preclinical models and the day of analysis in the T-cell transfer model.

For example, IL6 scale is between 0-400 pg/ml in DSS preclinical model and 0-800 in T-cell transfer preclinical model; IL13 is 0-200 pg/ml in DSS preclinical model and 0-800 pg/ml in T-cell transfer preclinical model; TNFa is 0-15 pg/ml and 0-600 pg/ml respectively, etc.

Author Response

Response to Reviewer 1 Comments:

Comments and Suggestions for Authors

Article has improved considerably. Authors have elaborated more focused discussion and conclusion sections together with the updating of the references

We thank the reviewer for their comments and feedback to help improve the manuscript and its clarity to readers. Based on your suggestions, we have merged the figures previously known as Figure 2 & Figure 3 into Figure 2A and 2B, and previously Figures 5 & 6 were merged into Figure 4A and 4B respectively. The captions and text were also modified to reflect these changes in the figures. In addition, the scales were made consistent across all 3 Figures within the context of the same inflammatory marker. We believe this will improve reader clarity going forward.

 Minor points:

  1. I insist on the joining of heatmap of Figure 2 with the cytokine graphs in Figure 3 in one Figure. By other side, I also insist on the joining of the heatmap on Figure 5 with the cytokine graphs in Figure 6 in the final version of the manuscript to make easier for the reader

          Figures were merged as suggested.

  1. I insist on the same scale sizes in the figure to be able to compare the cytokine levels between the preclinical models and the day of analysis in the T-cell transfer model.

For example, IL6 scale is between 0-400 pg/ml in DSS preclinical model and 0-800 in T-cell transfer preclinical model; IL13 is 0-200 pg/ml in DSS preclinical model and 0-800 pg/ml in T-cell transfer preclinical model; TNFa is 0-15 pg/ml and 0-600 pg/ml respectively, etc.

Thank you for clarifying this point, we misunderstood it previously. Scale sizes were made identical across all 3 sets of data. This will make referring across Figures much simpler for the reader.